# Application of Photopolymer Materials in Holographic Technologies

**DOI:** 10.3390/polym11122020

**Published:** 2019-12-06

**Authors:** Nadezhda Vorzobova, Pavel Sokolov

**Affiliations:** Department of Photonics and Optoinformatics, ITMO University, Kronverksky pr. 49, 197101 Saint Petersburg, Russia; sokol7858@yandex.ru

**Keywords:** photopolymer materials, holographic technologies, holographic optical elements, periodic structures, holographic 3D printing

## Abstract

The possibility of the application of acrylate compositions and Bayfol HX photopolymers in holographic technologies is considered. The holographic characteristics of materials, their advantages, and limitations in relation to the tasks of obtaining holographic elements based on periodic structures are given. The conditions for obtaining controlled two and multichannel diffraction beam splitters are determined with advantages in terms of the simplicity of the fabrication process. The diffraction and selective properties of volume and hybrid periodic structures by radiation incidence in a wide range of angles in three-dimensional space are investigated, and new properties are identified that are of interest for the development of elements of holographic solar concentrators with advantages in the material used and the range of incidence angles. A new application of polymer materials in a new method of holographic 3D printing for polymer objects with arbitrary shape fabrication based on the projection of a holographic image of the object into the volume of photopolymerizable material is proposed, the advantage of which, relative to additive 3D printing technologies, is the elimination of the sequential synthesis of a three-dimensional object. The factors determining the requirements for the material, fabrication conditions, and properties of three-dimensional objects are identified and investigated.

## 1. Introduction

The great interest in using photopolymer materials in holographic technologies is determined by a number of advantages relative to other holographic materials, such as silver halide materials, dichromated gelatin (DCG), and photorefractive glasses, associated with their relatively low cost and simplicity of the processes for elements fabrication that are formed as a result of polymerization under light exposure without laborious preliminary or subsequent processing. Applied to holographic technologies, the most widely considered in the literature are photopolymerizable acrylamide (AA/PVA) compositions [1,2], compositions with the introduction of various additives to obtain new properties, including liquid crystals [3,4] and inorganic nanoparticles [5,6,7], materials with a diffusion recording mechanism (PQ/PMMA) [8,9]. The processes in materials, their properties, as well as the properties of holographic elements for various practical applications, including data storage [10,11], sensors [12], protective elements [13], and elements of holographic solar concentrators [14,15] are studied.

Continuous improvement and development of new materials stimulates the development of existing and the emergence of new applications. However, despite the improvement and provision of new properties, many materials or their components are not sufficiently accessible, difficult to prepare, and the processes of obtaining, and properties of, elements do not sufficiently satisfy the increasing requirements of practical tasks. Thus, if such traditional areas as data storage and sensors are based on highly selective properties of elements that are well studied, then a relatively new task of fabrication elements of holographic solar concentrators requires high diffraction properties in combination with a wide angular range. This problem is currently not sufficiently resolved. Traditional task such as obtaining controlled beam splitters need to be developed, as well as brand protection and authentication elements that require constant updating. To solve a number of problems, it is preferable to use industrial materials, which determines the great interest in the new photopolymer Bayfol produced by Covestro AG (Bayer MaterialScience) [16]. 

In this paper, we will consider two types of promising polymer materials: acrylate polymerizable compositions, the advantage of which is the use of available commercial components and ease of fabrication, and Bayfol HX material, the advantage of which is mass production. We will consider the possibility of their use in holographic technologies in relation to the problems of obtaining elements based on periodic structures, including diffraction elements of holographic solar concentrators and light beam splitters in order to obtain new properties and develop existing directions. In addition, the possibility of a new application of materials in the new method of holographic 3D printing that we have proposed to obtain polymer objects of a three-dimensional configuration, the main advantage of which with respect to additive 3D printing technologies is the elimination of sequential synthesis of a three-dimensional product, will be considered. The method has not been practically studied and the aim of the work is to identify and study the factors that determine the requirements for fabrication conditions and properties of three-dimensional objects.

## 2. Materials and Methods 

The materials studied are photopolymerizable acrylate compositions developed at ITMO University [17] and Bayfol HX photopolymer. In this section, we consider the properties and characteristics of materials determined the possibility of their using in holographic technologies.

### 2.1. Acrylate Compositions

The main components of the acrylate compositions are bisphenol A glycerolate (Sigma-Aldrich, No. 41.116-7, St. Louis, MO, USA) and 2-carboxyethyl acrylate (Sigma-Aldrich, No. 552348, St. Louis, MO, USA) in a 3:7 ratio and 0.5 wt % Irgacure 784 (Ciba, Basel, Switzerland) polymerization initiator providing sensitivity in blue and green areas of the spectrum. We used a monomer composition based on these components and nanocomposite with the introduction of silicon oxide nanoparticles (Sigma-Aldrich, No. 066K0110, St. Louis, MO, USA) with a size of 14 nm and a concentration of 6 wt % providing maximum diffraction efficiency.

Considering the diffraction properties of acrylate compositions and their determining processes, the study of diffraction properties was carried out by recording transmission holographic gratings in two beams interference schemes using radiation of helium-cadmium laser with a wavelength of 0.44 μm. The frequency (period) of the structures was changed with a change in the angle between the interfering beams. Diffraction efficiency was determined as the ratio of the intensity in the first diffraction order to the incident radiation intensity at a wavelength of 0.65 μm with TE polarization of the probe beam. Note here that when changing the TE polarization on TM polarization, we did not reveal significant differences in the maximum values of diffraction efficiency.

The refractive index modulation in the holographic grating is determined by the known mechanism of diffusion mass transfer due to differences in the polymerization rates of the composition components [5,6]. This mechanism determines the non-monotonous character of the dependence of diffraction efficiency on the exposure duration in the studied materials [18]. 

Monomeric composition and nanocomposites with silicon oxide are not critical to subsequent exposure and at optimal recording frequencies and layer thicknesses provide diffraction efficiency of up to 80% (Figure 1). High diffraction characteristics determine the possibility of using such materials for diffraction beam splitters fabrication (Section 3.1), as well as in security printing technologies (Section 3.2).

The dependence of diffraction efficiency on the recording frequency in accordance with existing concepts is determined by the processes of nonlocal polymerization [19]. As was shown in our paper [18], these processes can lead to the formation of bridges between the polymerization regions at the initial stage of structure formation and to an increase in light scattering. However, note here that this result can be considered as positive and used to obtain unidirectional diffusers with an elongated (along the dielectric planes of the grating) scattering indicatrix. 

An important property of acrylate monomer composition is the possibility of formation hybrid structures, that is, volume gratings with refractive index modulation, on the surface of which surface relief gratings are formed. As was shown in our previous work [20] the surface relief is formed when the coating film is removed. Gratings with the highest relief height are formed in a narrow range of exposures at the initial stage of structure formation. The formation of a relief grating leads to an increase in the diffraction efficiency (Figure 2). The difference in diffraction efficiency of the hybrid structure and the volume grating determines the diffraction efficiency of the relief component. 

The formation of a surface relief leads to interesting properties that can be used in relation to the problem of obtaining diffraction elements of solar concentrators (Section 3.3).

The advantages of materials are: the use of available commercial components, ease to prepare, the ability to obtain high diffraction efficiency (up to 80%), application to various substrates, the ability to change the thickness of the layer and introduction of additives to change its properties. The disadvantages include laboratory synthesis and low resolution. In accordance with Figure 1a, holographic recording is possible up to frequencies of 2700 l/mm, but maximum diffraction efficiency is achieved in a narrow frequency range near 330 l/mm. 

### 2.2. Bayfol HX Photopolymers

The advantages of Bayfol HX photopolymers are: mass production, higher resolution compared to acrylate compositions, and a wide range of spectral sensitivity [16]. Such materials can be used to record reflective holograms reconstructed in white light, however, as our experiments have shown, the diffraction efficiency is inferior to the classic high-resolution silver-halide materials and DCG. Therefore, in this article we will consider the possibility of using Bayfol HX photopolymers to obtain holographic elements operating in transmission mode, in which the diffraction efficiency of at least 80% is achieved. Figure 3 shows the dependence of diffraction efficiency on exposure time obtained by us. The recording wavelength is 0.63 μm, the power density is 12 mW/cm^2^, the layer thickness is 16 μm, and the period of the structures is 1.6 μm. 

## 3. Results and Discussion

In this section, we consider the possible applications of materials in holographic technologies as applied to the problems of obtaining elements based on periodic structures and elements of an arbitrary three-dimensional configuration, and also discuss their properties and possible applications.

### 3.1. Diffraction Beam Splitters

High diffraction characteristics determine the possibility of using acrylate compositions to obtain diffraction beam splitters while simplifying the fabrication processes relatively classical holographic materials. Figure 4 shows examples of splitters based on one-dimensional and two-dimensional structures. The structures were recorded in a nanocomposite with silicon oxide at a wavelength of 0.44 μm. One-dimensional structures were obtained with the exposure duration of 3 min at a power density of 2.6 mW/cm^2^. To fixing unexposed material, we used subsequent exposure to UV radiation from a mercury lamp with a power density of 10 W/m^2^ for about 10 min. A photograph of the structure (Figure 4a) was obtained using an OLYMPUS STM6 optical microscope (OLYMPUS, Waltham, MA, USA). Two-dimensional structures were obtained by sequentially recording two gratings when the sample was rotated by 90° for the second exposure. The exposure duration was optimized by changing the exposure components from 1 to 4 min. The best result was obtained with durations of the first and second exposures of 3 min. Subsequent treatment in isopropanol (about 10 min) was used to remove the unpolymerized material. A photograph of the structure (Figure 4d) was obtained using a Labomed-3 optical microscope (Labomed, Los Angeles, CA, USA). Some cells appear smeared, as we assume, due to residual material after treatment in isopropanol, or nonlocal polymerization processes. The diffraction pattern on the structure is shown in Figure 4e (some distortions are associated with the camera angle and the use of slightly diverging beams for recording).

The possibility of changing the layer thickness and the period of the structures makes it possible to obtain both two-beams splitters based on volume gratings and multi-beams splitters based on thin gratings. Elements have high light, moisture, and heat resistance. It should be noted, however, that volume gratings have high angular selectivity. The half-width of the angular selectivity contour is less than 2°, which makes high demands on the accuracy of positioning of the beam splitters to provide the required intensities ratio at the output of the element.

In this regard, we investigated the possibility of obtaining two and multichannel splitters in which a change in the radiation intensity in the channels can be achieved by rotating the element without presenting high requirements for positioning accuracy. Two-beam splitters were recorded in Bayfol HX material by the interference of two beams and optimal exposure. Higher resolution relative to acrylate compositions allows us to record beam splitters with a greater range of angles at the output of the element.

The structures of multi-beams splitters were recorded in the Bayfol HX material by the method of interference copying (at wavelength of 0.63 μm) of template structures—two-dimensional gratings (with a period of 1.6 μm). To obtain template (matrix) structures, we used the industrial silver halide material PFG-03M (Slavich, Pereslavl-Zalessky, Russia). The use of such fine-grained material for the template is associated with lower light scattering compared to polymers and less criticality to sequential recording, as well as with glass substrate. During interference copying, the Bayfol HX film was placed behind the template. Exposure was carried out by a parallel laser beam for 10 s (at a power density of 12 mW/cm^2^). The advantage of the interference copying compared to direct recording on Bayfol HX films is the ability of elements replication by reducing the requirements for recording stability (vibration isolation), laser output power, and the simplicity of the recording scheme.

The combination of high diffraction properties and the resulting angular selectivity (Figure 5) make it possible to change the radiation intensities in diffracted beams when the element is rotated without high demands to the accuracy of the angular displacements.

Figure 6 shows the diffraction patterns on a five beams splitter.

The elements have advantages over the elements in classical holographic materials associated with the material used and the simplicity of the fabrication process, which does not require laborious preliminary or subsequent processing. Such diffraction splitters can be used and are used by us in various schemes of a physical experiment, in particular in holographic recording schemes requiring the separation of a laser beam into two or more beams with adjustable intensity. We also assume the possibility of using controlled multi-channel splitters in secure printing technologies when information is input into interfering beams, which can provide several security features that may recognized by visual and instrument control. It is also possible to use elements as multiplexers for various tasks, including introducing radiation into optical fibers, with the advantage in the simplicity of the process of their fabrication relative to lithography methods [21].

### 3.2. Protection Technologies

The low resolution does not allow the recording of highly efficient reflective holograms reconstructed in white light in the studied materials. For acrylate compositions, the operating frequency range is up to 2700 l/mm (Figure 1a) however, diffraction efficiency is reduced to units of percent at high frequencies. The resolution of Bayfol HX according to the developer [16] is about 7000 l/mm, but as our experiments have shown, when recording reflection holograms (at wavelength of 0.63 and 0.53 μm), the diffraction efficiency is reduced to 30%. However, such materials, as were shown in our previous work, can be used to obtain holograms of a focused image that do not require high resolution. We have fabricated elements based on holograms of a focused image, as well as elements based on volume transmission gratings, with reconstruction of images in white light [18]. The film-based elements were recorded in acrylate nanocomposite with silicon oxide. Elements have high brightness, as well as light, moisture, and heat resistance (up to 150 °C), which determines the possibility of using such elements in brand protection technology. Similar elements can also be obtained in film Bayfol HX material with a decrease in angular selectivity (Figure 5b), which is important for visual authentication. 

### 3.3. Elements Based on Periodic Structures for Energy Application

The interest in the problem of obtaining holographic solar concentrators in photopolymer materials is determined by the fact that such elements can be obtained by relatively simple methods, they are light, compact, and cheap. The main requirement for holographic solar concentrators is a combination of high diffraction efficiency with a wide range of angles of radiation incidence. In the literature are considered the processes of fabrication and properties of diffraction deflectors based on volume [22,23,24,25] or relief structures [26] and also focusing elements [27,28,29,30] as components of solar concentrators. In this section, we consider the properties of structures in Bayfol HX materials and acrylate compositions as applied to the problem of obtaining diffraction deflectors of holographic solar concentrators.

First, we consider the properties of volume gratings in Bayfol HX photopolymer when radiation is incident over a wide range of angles in three-dimensional space. Figure 7a shows the dependence of the diffraction efficiency of volume transmission non-slanted grating (with a period of 1.6 μm) on the angles α and β (Figure 7b) corresponding to the movement of the sun during the day and the seasonal change in height above the horizon. Diffraction efficiency measurements were carried out at a wavelength of 0.65 μm.

You can see many directions of radiation incidence on the grating (values of angles α and β) at which the maximum values of diffraction efficiency are achieved. It should be noted that when the direction of incidence changes in the Bragg plane (the plane slanted at a Bragg angle (ϴ_B_) to the normal to the grating surface), the maximum diffraction efficiency (about 80%) remains in the range of angles α over 100°. When deviating from the Bragg plane, i.e., an increase in the angle β (up to 30°), the diffraction efficiency maximum shifts toward greater angles α. This result is important and solves the problem of solar radiation using at great angles of incidence, up to 70°.

Let us now consider the properties of the structures recorded in the acrylate monomer composition. The advantage of this composition is the possibility of a hybrid structure formation, i.e., a volume grating with refractive index modulation, on the surface of which a relief grating is formed. It should be noted that in the Bayfol HX material we were not able to obtain a surface relief, which may be associated with the mechanism of its formation, which is mostly manifested in the liquid composition.

As was shown in our previous work [20], the addition of a surface relief leads to a substantial broadening of the angular selectivity contour of the hybrid structure as compared to the volume grating. However, this result was obtained for radiation incidence in one plane. In this paper, we present the results of a study of the diffraction properties of hybrid structures when radiation is incident in a wide range of angles in three-dimensional space (Figure 8).

One can see the advantages of hybrid structures compared to volume structures in Bayfol HX over the general range of incidence angles, in particular, β angles, determined by the contribution of the relief component of the hybrid structure. An increase in the range of angles β means the possibility of using hybrid structures without tracking the trajectory of the sun, not only during the day, but also with a significant change in its height above the horizon during the year. In addition, the established properties determine the advantages of using such structures in mobile devices. 

Thus, the established properties of volume and hybrid structures determine the possibility of development diffraction deflectors of holographic solar concentrators that redirect radiation in a wide range of angles in one direction. The established ranges of incidence angles, in which high diffraction efficiency is maintained, exceed the ranges presented in [22,23,24,25,26]. Given the possibility of using solar radiation passing through the gratings (outside the range of angles corresponding to the diffraction maximum), the general ranges are: for angles α about 140° and angles β up to 60° (for volume structures) and more than 120° (for hybrid structures). Such elements can be used independently, as well as in combination with focusing elements, which will reduce the working area of solar cells. 

### 3.4. Holographic 3D Printing 

Consider the new method proposed by us in our previous works [31,32] for fabrication objects with an arbitrary three-dimensional shape, which we will call the method of holographic 3D printing. The idea of the method is to project a holographic image of the formed object into the volume of the photopolymerizable material and display it in the material with the formation of a three-dimensional object. Figure 9 shows a scheme of holographic 3D printing. The beam of laser (1) is expanded (and filtered if necessary) by the optical system and sent on the lens (2), which forms a converging beam to obtain a real image of the object formed by the hologram (4). To reduce the size of the holographic image, an optical system (5) is introduced into the diffracted beam, which projects the image of the object into the volume of the photopolymerizable material (6). (7) is the reduced real image of the formed object.

The advantage of the method relatively to additive 3D printing technologies based on point-by-point and layer-by-layer formation of a three-dimensional object is the exclusion of sequential synthesis. The total object is formed as a result of a single light exposure of the material.

To display the configuration of an object, it is necessary to polymerize the material in the region of greatest sharpness of the holographic image. However, in the forming beam, before and beyond this region, the material can also polymerize. Therefore, it is necessary to limit the polymerization outside the region of greatest sharpness. We propose the following principles for limiting the polymerization:Use of materials with a radical polymerization and oxygen inhibition of the process. Oxygen contained in the air can enter through the upper surface of the material (it can also be forced) and limit the polymerization mainly in the upper part of the layer.Creating an appropriate absorption of the forming radiation in the material, for example, with the introduction of absorbent additives. In this case, the polymerization is limited mainly in the lower part of the layer.An increase in the radiation intensity gradient in the beam forming the reconstructed image that can be provided by the objective.

Without limitation of polymerization the layer polymerizes over its entire thickness (Figure 10b). However, a gradient object can be formed inside the layer as a result of the dependence of the refractive index on exposure (Figure 10c). The refractive index modulation was calculated using the well-known formula given in particular in [20] and the dependence of diffraction efficiency on the exposure shown in Figure 2.

Figure 11 and Figure 12 show the results of experiments on the implementation of the holographic 3D printing method with limitation of polymerization [31]. The projecting transmission holograms were recorded by pulse radiation of single-pulse laser with a wavelength of 0.53 μm, output energy of 1 J, and pulse duration of 10 ns. The forming radiation was the radiation of a DPSS laser with a wavelength of 0.53 μm and output power of 200 mW. The recording material was a monomeric acrylate composition. Figure 11 shows a view of a reconstructed image, a view of a polymer object on glass and separated from the substrate. Figure 12 shows a field of model objects with flat, spherical, slanted, and stepped surfaces and a field of polymer elements with a surface shape corresponding to model surfaces.

The method is new and practically unexplored. There are many questions related to factors that determine the requirements for the formation conditions and properties of the formed objects. We will not dwell on the results of our studies that are not related to the topics of this journal. In this paper, we consider the factors that determine the requirements for the material and exposure parameters.

The relationship of material parameters and exposure parameters is determined by expressions:(1)$$I_{\mathrm{im}}e^{-k_{\lambda }T_{\mathrm{im}}}t_{\exp}\geq E_{0}$$
(2)$$I_{\mathrm{im}}=\frac{P\eta \alpha }{S}$$
(3)$$t_{exp}\geq \frac{E_{0}Se^{k_{\lambda }T_{\mathrm{im}}}}{P\eta \alpha }$$
where *I*_im_ is the radiation intensity in the reconstructed image (excluding absorption in the layer), *k*_λ_ is the absorption coefficient of the material, *T*_im_ is the thickness of the layer from its surface to the area of localization of the image, *t_exp_* is the exposure duration, and *E*_0_ is the threshold value of polymerization energy, *P* is the output power of laser, Ƞ is the diffraction efficiency of the hologram, α is the loss coefficient in the recording scheme, and *S* is the area of the formed object.

Using Equation (3), the graphs shown in Figure 13 were obtained. The obtained dependences make it possible to determine the requirements for the absorption of the forming radiation in the material (with the aim of limiting the polymerization beyond the region of greatest sharpness of the projected image), exposure time, localization of the projected image in the volume of material and the profile depth of the formed three-dimensional object.

The curves show, in particular, that high absorption in the layer, which can completely eliminate polymerization beyond the region of greatest sharpness of the holographic image, leads to an increase in the exposure time. The exposure duration depends on the length of the object in depth (depth of the object profile). For the formation of object fragments located at different distances from the surface of the layer, different exposure times are required. The optimal time should be determined by the exposure time for fragments farthest from the surface. However, too long a duration, determined by the lower fragment, can lead to distortion of the shape of an object fragment located closer to the layer surface, which must be taken into account when implementing the method.

The absorption in the photopolymer material is determined by the initiating system (polymerization initiators, sensitizers, their concentration), as well as the wavelength of the forming radiation. The required absorption can also be created by the introduction of additional additives. For example, the object shown in Figure 11c was obtained when Rhodamine B dye was introduced into the material.

The possibility of implementing the method has been experimentally confirmed, a number of factors have been identified and investigated that determine the requirements for the material and the conditions for obtaining polymer objects, however, there are some questions that we intend to consider in subsequent work. The method of holographic 3D printing can be used in technique, including small-sized elements fabrication, medicine, including tissue engineering, as well as a new type of art graphics.

## 4. Conclusions

The analysis of the properties and characteristics of two types of photopolymer materials—photopolymerizable acrylate compositions and industrial Bayfol HX photopolymers—determining the possibility of their application in holographic technologies has been carried out. The advantages and limitations of the use of such materials applied to the problems of obtaining elements based on periodic structures and elements with an arbitrary three-dimensional configuration have been considered. The possibility of obtaining controlled light beam splitters has been shown, with advantages in the simplicity of the fabrication process, as well as in the physical properties of the elements and their operational characteristics. The diffraction and selective properties of volume and hybrid periodic structures by radiation incidence in a wide range of angles in three-dimensional space have been investigated. New properties have been identified that are of interest for the development of elements of holographic solar concentrators with advantages in the material used and the range of incidence angles which exceeds the ranges presented in previous works. New application of polymeric materials has been proposed in a new method of holographic 3D printing, which can be an alternative to additive technologies with the advantage of eliminating the sequential synthesis of a three-dimensional object. The all object is formed as a result of a single exposure of the material. The factors determining the requirements for the material, formation conditions, and the properties of three-dimensional objects have been investigated. The requirements for the absorption of the forming radiation in the material, exposure time, localization of the projected image in the volume of material and the profile depth of the formed three-dimensional object have been determined. The possibility of obtaining gradient objects and three-dimensional polymer objects with an arbitrary surface shape has been shown.

## Figures and Tables

**Figure 1 polymers-11-02020-f001:**
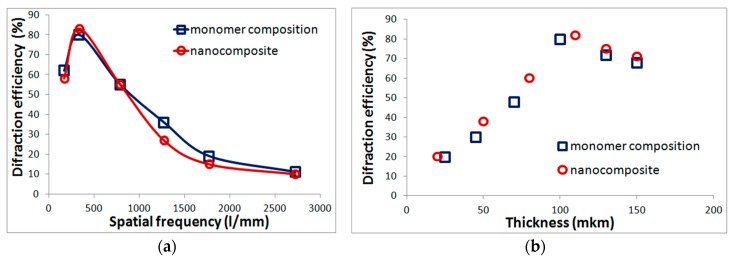
The dependence of diffraction efficiency on the recording frequency (**a**) and layer thickness (**b**) for the monomer composition and the nanocomposite with silicon oxide.

**Figure 2 polymers-11-02020-f002:**
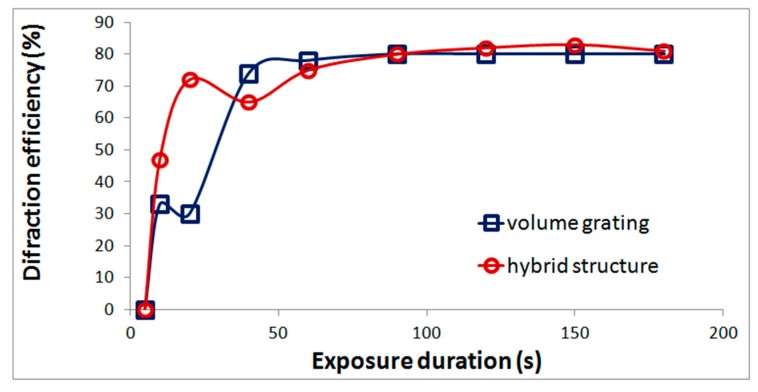
Dependence of diffraction efficiency on exposure duration for a volume grating and a hybrid structure recorded in the monomer composition.

**Figure 3 polymers-11-02020-f003:**
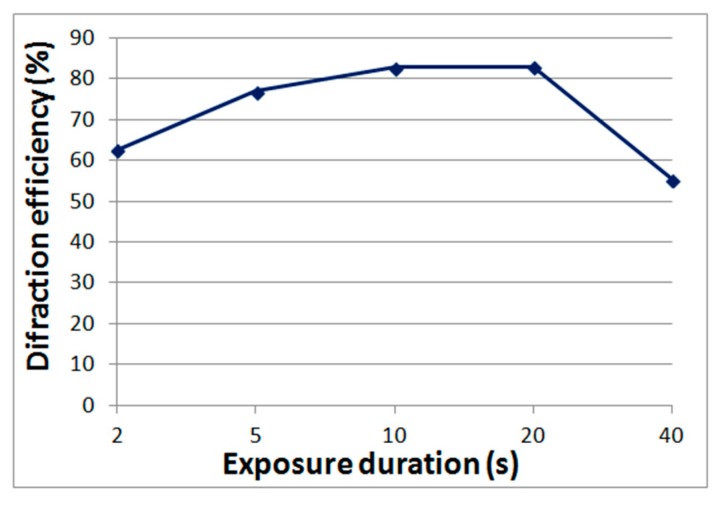
The dependence of the diffraction efficiency on the exposure duration for the Bayfol HX photopolymer.

**Figure 4 polymers-11-02020-f004:**
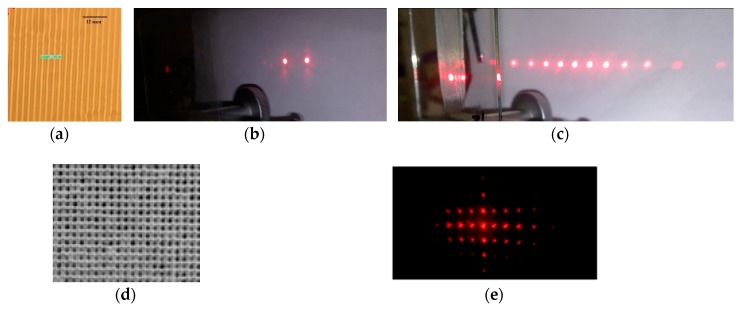
Examples of structures of two and multichannel splitters, recorded in the nanocomposite with silicon oxide and diffraction patterns. (**a**,**b**) Structure of a volume one-dimensional grating and diffraction pattern (the period is 3 μm, thickness is 100 μm); (**c**) Diffraction pattern on a thin one-dimensional grating (the period is 6 μm, thickness is 20 μm); (**d**,**e**) Structure of a thin two-dimensional grating and diffraction pattern.

**Figure 5 polymers-11-02020-f005:**
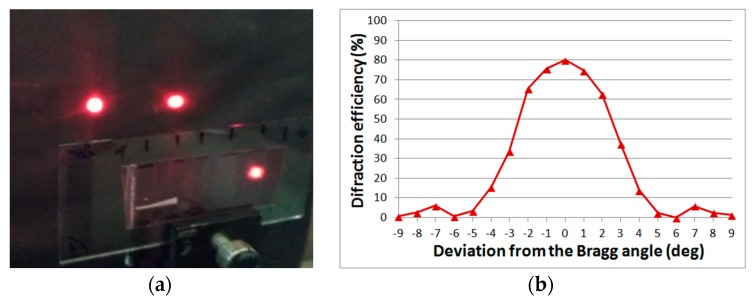
The diffraction pattern on a two-beam splitter in Bayfol HX material (**a**) and the angular selectivity contour (**b**).

**Figure 6 polymers-11-02020-f006:**
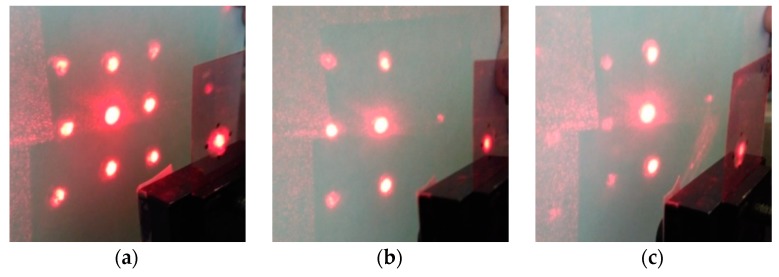
(**a**) The diffraction pattern on a five beams splitter, the ratio of intensities I_+1_/I_0_ is 1: 1.5, (**b**,**c**) Change in the diffraction pattern when the element is rotated on 6° (**b**) and 25° (**c**).

**Figure 7 polymers-11-02020-f007:**
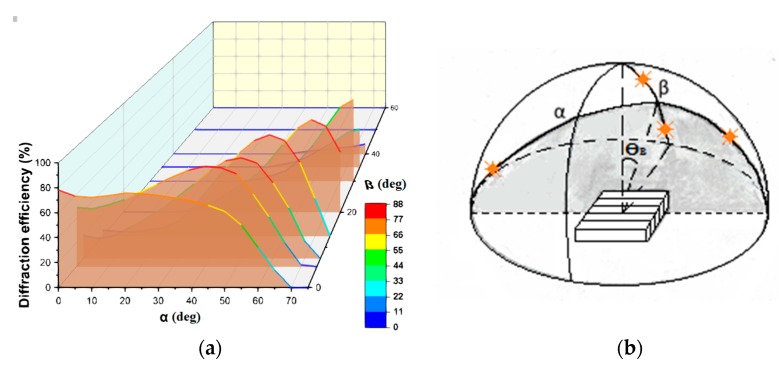
(**a**) The dependence of the diffraction efficiency of the volume grating recorded in the Bayfol HX on the angles α and β; (**b**) scheme explaining the grating position and variable angles of incidence.

**Figure 8 polymers-11-02020-f008:**
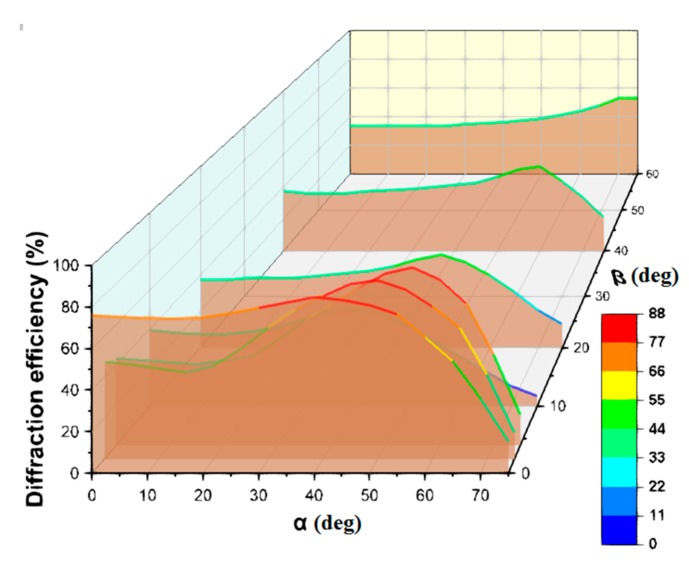
The dependence of the diffraction efficiency of hybrid structures recorded in the acrylate monomer composition on the angles α and β.

**Figure 9 polymers-11-02020-f009:**
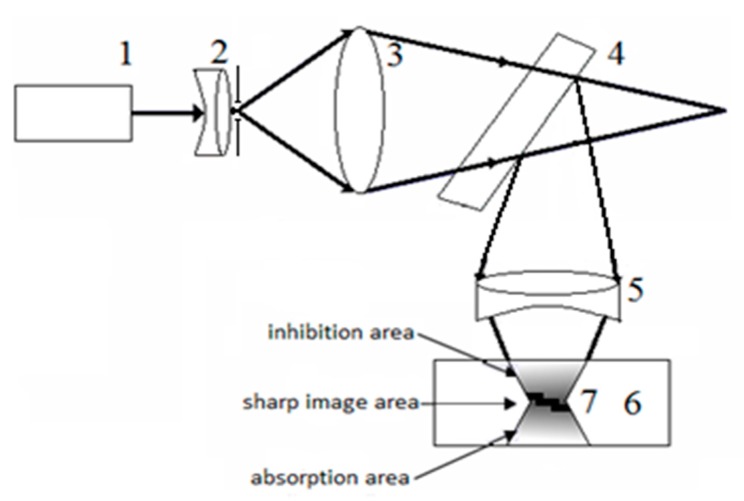
Scheme of holographic 3D printing with reducing the size of the projected image. (**1**) laser, (**2**,**3**) optical system forming the convergent beam, (**4**) hologram, (**5**) reducing optical system, (**6**) photopolymerizable material, and (**7**) holographic real image.

**Figure 10 polymers-11-02020-f010:**
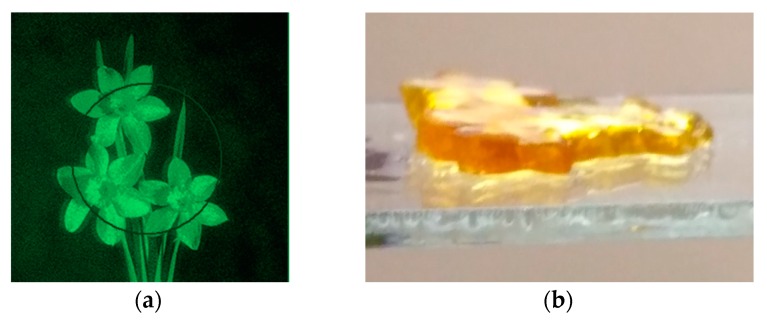
View of the holographic image (**a**) and the object obtained without limiting of polymerization (**b**). View of the gradient object (**c**) obtained as a result of the dependence of the refractive index on exposure (**d**).

**Figure 11 polymers-11-02020-f011:**
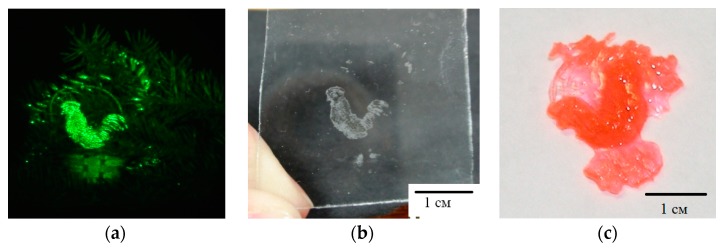
View of a holographic image (**a**), a polymeric object on a glass substrate (**b**) and separated from the substrate (**c**).

**Figure 12 polymers-11-02020-f012:**
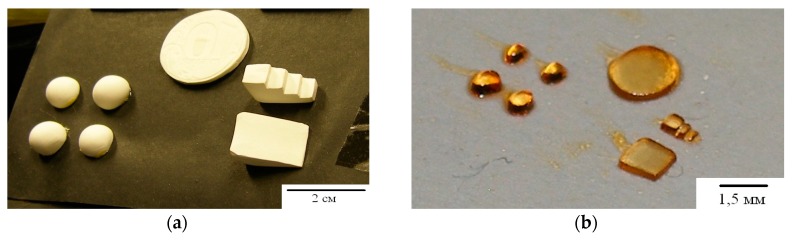
The field of model (**a**) and polymer (**b**) objects [31].

**Figure 13 polymers-11-02020-f013:**
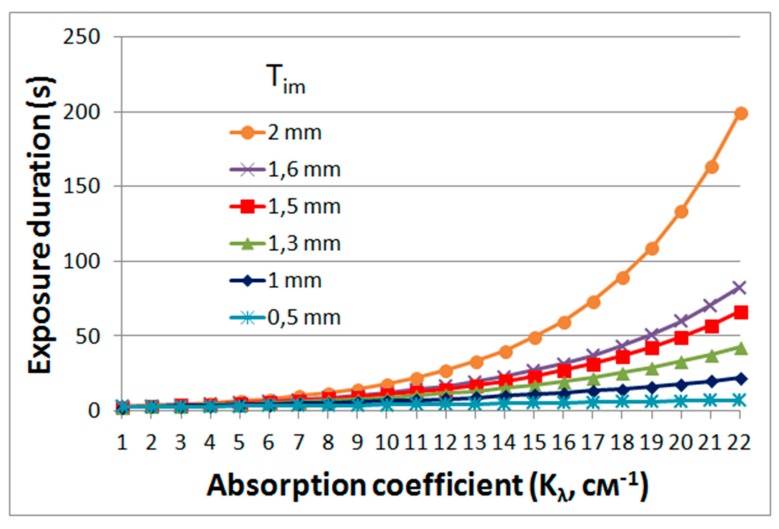
Dependence of exposure duration on absorption coefficient (*k*_λ_) of the material and localization of the projected holographic image (*T*_im_). *T*_im_ = 0.5–2 mm, Ƞ = 20%, *P* = 200 mW, α = 0.5, *S* = 0.5 cm^2^, *E*_0_ = 0.1 J/cm^2^.

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
