# Peer review of "Application of Photopolymer Materials in Holographic Technologies"

_polymers, 2019, doi:10.3390/polym11122020_

Round 1

Reviewer 1 Report

In this paper, the authors consider the possibility of the application of two different photopolymers in holographic technologies. They also study and show the improvement in holographic characteristics of nanocomposite material with the introduction of nanoparticles of silicon oxide.

Continuous improvement and development of new materials as well as the development of diffractive optical elements is important for the emergence of new applications in the field.

The reported results are relevant and important. However, the manuscript needs some clarifications and additions. Here are several remarks (not listed in order of importance), which have to be considered:

The abstract is too short and general. It needs to be supplemented a bit to be more informative and specific. Some points from the materials and methods section should be explained more explicitly. Such as the size of nanoparticles and why these concentatrations were chosen? Line 51. In the whole work, properties studied and applications are in materials of nanocomposite with silicon oxide and Bayfol HX photopolymer. There are no studies or applications of zinc oxide nanocomposites. For this reason, I recommend removing this material from the section on materials used in line 51. In section results and discussion in lines 120-121:”optimal exposure parameters and subsequent exposure to UV radiation”. What are the optimal exposure parameters for both the 442nm and UV radiation, which were used to record the structures shown in Figure 4? It should be mentioned what the pictures are in figure 4 a) and d)? AFM and STEM? How is the two-dimensional diffraction grating recorded, as in the case shown in Figure 6 by copying the interference or by recording two cross grating? What is the exposure of each of them? Lines 142-144. Authors should explain or reference the method of copying. What is the advantage of the Bayfol HX material of the matrix material, why not make the element directly in the matrix material? At what angle is the element in pictures b) and c) in Figure 6 rotated? Lines 142-144. “The low resolution does not allow the recording of highly efficient reflective holograms reconstructed in white light in the studied materials”. What is the resolution of the materials used? Figure 7 b) is unclear and low resolution. It will be useful, if its quality can be improved.

Author Response

Dear reviewer,

The authors thank you for reviewing manuscript, as well as for comments and recommendations. We took into account all the comments and recommendations and made the appropriate changes and additions to the text and figures. Modified version of the manuscript and responses to comments are in the attached file.

Reviewer 2 Report

At first glance, the manuscript is interesting for this special issue. It analyses two interesting holographic application using two different photopolymers acrylate compositions and Bayfol HX.  Nevertheless, in my opinion, some parts of the paper must be improved.

Experimental data figures are so poor designed, please improve this and difference which point are experimental, how they interpolate the line in figure 1.a for example? Usually the units of the axis are between parentheses. They should define DE, I assume they obtain de DE comparing the diffracted intensity with the incident one, then it is important the polarization of the incident beam because the Fresnel losses are different for TE and TM polarizations. In other words the maximum DE depends on the prove beam polarization. Figure 2. Are real time measurements? Which points are experimental and which one interpolated? Line 96 explain better the sentence “disadvantages include low resolution and laboratory synthesis” provide if it possible numerical data of the resolution

Explain better how is generated the recording pattern of Figure 4.d. Some of the cells look different. Why the spots of Fig. 4.e seem shifted, is no symmetrical using vertical line. I assume that are non-slanted the gratings recorded. 5.b The maximum DE is just 1%? In fig. 5.a seems higher. The angular scan need additional experimental measurements to see the symmetry and the secondary lobes better. The affirmations between lines 159 and 162 need to be supported by references.

Line 268: the authors claim that they use pulsed radiation; therefore the characteristics of this laser should be given for the readers. Energy and frequency of the pulse. P is the output power of laser, in general, in holography the laser beam is filtered and expanded, and then the authors give the mW/cm2 Line 314, there isn’t any object shown in Fig. 13, maybe the authors mention the object from fig. 10?

In general the original results provided in this paper should be emphasizes better from the provided in previous papers.

Some time is confused for a reader know with results are provided with Bayfol and which one with the acrylates photopolymer, I recommend to introduce this in figure captions for example.

To summarize, the paper should improved and clarified before acceptance. If the authors do not improve all points listed this paper cannot be accepted. 

Author Response

(The authors gave the same response as above.)

Round 2

Reviewer 2 Report

The manuscript has been clearly improved by the authors.